# The Truth, The Whole Truth, and Nothing but the Truth:
# A New Benchmark Dataset for Hebrew Text Credibility Assessment

**Ben Hagag**
Bar-Ilan University
ben.hagag@biu.ac.il

**Reut Tsarfaty**
Bar-Ilan University
reut.tsarfaty@biu.ac.il

## Abstract

In the age of information overload, it is more important than ever to discern fact from fiction. From the internet to traditional media, we are constantly confronted with a deluge of information, much of which comes from politicians and other public figures who wield significant influence. In this paper, we introduce HeTrue: a new, publicly available dataset for evaluating the credibility of statements made by Israeli public figures and politicians. This dataset consists of 1021 statements, manually annotated by Israeli professional journalists, for their credibility status. Using this corpus, we set out to assess whether the credibility of statements can be predicted based on the text alone. To establish a baseline, we compare text-only methods with others using additional data like metadata, context, and evidence. Furthermore, we develop several credibility assessment models, including a feature-based model that utilizes linguistic features, and state-of-the-art transformer-based models with contextualized embeddings from a pre-trained encoder. Empirical results demonstrate improved performance when models integrate statement and context, outperforming those relying on the statement text alone. Our best model, which also integrates evidence, achieves a 48.3 F1 Score, suggesting that HeTrue is a challenging benchmark, calling for further work on this task.

## 1   Introduction

Our society is struggling with an unprecedented amount of falsehoods, hyperboles, and half-truths (Hassan et al., 2017). False information, commonly termed 'fake news', is now viewed as one of the greatest threats to democracy, journalism, and freedom of expression. Distributing false content has become a significant concern in recent years when it is claimed to have helped change public opinion in the US elections of 2016 (Zhou and Zafarani, 2018). Our economies are not immune to the spread of fake news either, with fake news being connected to stock market fluctuations and large trades. For example, fake news claiming that Barack Obama, the 44th President of the United States, was injured in an explosion, wiped out $130 billion in stock value (Zhou and Zafarani, 2018). Also, recent studies have shown that false content reaches "farther" and a wider audience of readers than real content (Vosoughi et al., 2018).

Even if a correction of the false information reaches the misinformed audience, simply providing the correct information is ineffective, as continued reliance on misinformation is likely when the misinformation conforms to a person's pre-existing belief system, yet the correction does not (Lewandowsky et al., 2005). Retracting misinformation that runs counter to a person's worldview can ironically even strengthen the to-be-corrected information, a phenomenon known as the worldview backfire effect (Hart and Nisbet, 2012).

Addressing the spread of fake news necessitates a comprehensive approach to assess the credibility of information. Credibility assessment, distinct yet sometimes used interchangeably with fake news detection, involves evaluating the veracity of statements or claims made by individuals, particularly those in influential positions. While fake news explicitly entails deception, credibility assessment encompasses a broader spectrum, scrutinizing the authenticity and reliability of information regardless of the intent behind it.

Manual credibility assessment is not sufficient in effectively combating the rapid spread of false information, leading to a pressing need for alternative approaches. In particular, there is a need for automatic credibility assessment tools, which leverages statistical and machine learning methods to analyze and determine the credibility of statements or news as they come in, thereby avoiding the human-labor bottleneck.

To effectively combat fake news and assess the

credibility of information, there is a pressing need for comprehensive datasets that facilitate the development and evaluation of automatic detection tools. Existing datasets primarily focus on English language content (Guo et al., 2022), leaving a substantial gap in the study of other languages, including Hebrew, which known for its unique linguistic characteristics (Tsarfaty et al., 2019). Additionally, many available datasets lack crucial supplementary information such as context, semantic features, and metadata associated with the statements in question. Including these features is essential for building robust NLP models, enhancing their understanding and accuracy in credibility assessment tasks.

The need for such fake-news detectors is particularly pressing in areas of which socio-political situations may be sensitive yet may have world-wide effects — such as the Middle East. To mitigate this, we introduce the *HeTrue* dataset, the first-ever Hebrew fake-news detection benchmark.

The *HeTrue* dataset is a unique Hebrew resource composed of 1021 statements made by politicians. Each statement is meticulously labeled for truthfulness and supplemented with context information. The dataset is further augmented with metadata and semantic features, which have been validated and annotated by nonpartisan fact-checkers, compiled with the International Fact-Checking Network (IFCN). While the HeTrue dataset addresses a significant gap in Hebrew language resources for credibility assessment, its comprehensive set of features also raises the bar for similar datasets in other languages, including English.

Utilizing this novel benchmark, in this paper we focus mainly on linguistic-based approaches for automatic credibility assessment — that is, one that relies solely on the claim, or the claim's context as input for detection. Linguistic approaches are frequently utilized to tackle this task (Wang, 2017, Roy et al., 2018), as they can operate without additional resources or data. This makes these approaches more scalable and straightforward to implement in real-time applications with minimal effort and budget.

These approaches hinge on the assumption that individuals tend to express themselves differently, whether verbally or in writing, when conveying false information compared to true information. Fraser (1991) asserts that these differences stem from a sense of stress, manifesting in a decrease in cognitive integration capacity, precision, organiza-

tion, and prioritization. These challenges result in a change in the normal elements of the speaker's language. Prior works have found that these linguistic signals can also specifically assist in distinguishing between truth and falsehood in politicians' statements (Wang, 2017).

We hypothesize that linguistic signals indicative of a statement's credibility exist in Hebrew and that these signals correlate with the credibility assessments made by professional journalists, who collaborated with us to create our gold-standard benchmark. To provide a broader view, we have also explored alternative methods, which will be discussed in the following sections.

As Hassan et al. (2015) first introduced, achieving the 'Holy Grail' in end-to-end credibility assessment systems, also referred to as fact-checking, requires a fully automatic platform. This platform should consist of three main components: a Claim-Spotter to detect claims that need validation in real time, a Claim-Checker to evaluate the veracity of these claims, and a Fact-Check Reporter to justify the evaluations with convincing evidence. This system would make determinations by analyzing the claim's text, source, claimant profile, and retrieving evidence from databases of previously checked claims. Our work advances towards this "Holy Grail," focusing on a fully automated credibility assessment system for Hebrew texts. The unique features of our dataset[1] and our experiments could also benefit credibility assessment in other languages. We call on the computing and journalism communities to join in on this pursuit, given the importance and timeliness of this task.

The contribution of our paper is, hence, manifold. First, we introduce *HeTrue*, the first publicly available Hebrew dataset for fake news detection. This dataset comprises 1021 statements from politicians and public figures, manually labeled by professional journalists, and serves as a valuable resource for researchers investigating fake news detection in Hebrew. To the best of our knowledge, this is the first claim assessment dataset that accompanies each claim with its context, alongside semantic and metadata features.

Second, we conduct an extensive experimental analysis, exploring a variety of modeling types and dataset setups. To establish a baseline performance for the task, we employ several methods including a hand-crafted feature-based model, a recurrent neu-

---

[1]See comparison with other datasets in Appendix A.7

ral network (RNN) with static embeddings, and a model extending AlephBERT—a Hebrew monolingual pre-trained encoder (Seker et al., 2021). Our experiments with these models demonstrate the presence of linguistic cues in Hebrew text that can distinguish between credible and non-credible statements. we provide an analysis of the key features from the hand-crafted feature model, emphasizing their significance.

Thirdly, we demonstrate the importance of integrating context and evidence in credibility assessment. Our novel model, which combines statement context and an evidence-retrieval mechanism, results in significant performance enhancements. We further compare our results with metadata-based models, revealing that our linguistic approach outperforms these models, and demonstrating that the addition of metadata features to a linguistic (handcrafted feature model) does not have an additive effect on the model's performance [2].

## 2 Related work

Credibility assessment is an interdisciplinary field that involves evaluating the believability, trustworthiness, reliability, accuracy, fairness, and objectivity of claims or statements (Viviani and Pasi, 2017). It is closely related to fact checking and deception detection, and focuses on assessing the quality of information and the level of trust that can be placed with respect to claims. The goal of credibility assessment is to differentiate between credible and non-credible claims, without necessarily determining the root cause of false information, such as whether it is due to incorrect factual claims or the intent of the speaker (Giachanou et al., 2019).

There are three categories of approaches for credibility assessment and fake news detection: linguistic (text-based), evidence-retrieval, and metadatabased, which typically incorporate machine learning techniques for training effective and accurate classifiers (de Souza et al., 2020).

Linguistic approaches attempt to identify clues in the text in order to verify veracity (Conroy et al., 2015, Zhou et al., 2004). The work of Afroz et al. (2012) posits that changes in certain linguistic characteristics can signal an attempt to conceal writing style, aiding in the detection of deceptive texts. This research present the effectivness of specialized set of lying-detection features including quantity

metrics (such as syllable and word counts), vocabulary and grammatical complexity, and an analysis of the usage of words denoting uncertainty, specificity, and expressiveness. On the other hand, Hancock et al. (2007) focus on linguistic traits more frequently associated with deceptive behavior. Their examination of 242 transcripts revealed that individuals inclined to deception tend to produce lengthier texts, utilize a greater number of sensory expressions (such as references to sight or touch), and exhibit a preference for using pronouns that shift focus away from themselves and towards others, in comparison to when they are being truthful. Reis et al. (2019) incorporates various psycholinguistic features derived from LIWC (Pennebaker et al., 2001), to detect persuasive and biased language. In our work, we systematically adapt and extend these methodologies to assess statement credibility in Hebrew.

The second approach is Evidence retrieval, which aims to find sources supporting or refuting the claim. One strategy involves the construction of knowledge graphs, facilitating the identification of relationships between textual segments and graph instances Pan et al. (2018). Alternatively, preestablished knowledge bases can be employed to garner pertinent evidence, subsequently leveraging models that capitalize on this retrieved information for text classification Thorne and Vlachos (2018).

The third approach is metadata-based, and it is often used alongside other methods. It involves analyzing text-related information like publication date, claimant's demographics, and source type (Long et al., 2017). Additionally, with the rise of fake news on social networks (de Souza et al., 2020), behavioral analysis becomes important to identify unreliable patterns through user or message interactions (Ruchansky et al., 2017).

In terms of the credibility levels to be predicted, the work by Rashkin et al. (2017) was first to introduce the task of credibility assessment in scale of 1-6, in their work with the Liar dataset (Wang, 2017). They used LSTM with GLOVE word embedding and showed that stylistic cues can help determine the truthfulness of text. Roy et al. (2018) applied CNN and BiLSTM models to the same dataset and task, aiming to extract patterns from short statements and understand the unique behaviors of source speakers using different dataset attributes, underscoring the importance of including speakers' profile information for fake news classifi-

---

[2]The HeTrue dataset and models are publicly available at: https://github.com/OnlpLab/HeTrue.

cation.

Giachanou et al. (2019) incorporated emotional signals for credibility assessment, which presented improved performance, relying on the fact that those signals can play a vital role in detecting false information (Ekman, 1992 Vosoughi et al., 2018).

Recent progress in this field encapsulates a variety of methodologies. Victor (2020), for instance, present a semi-supervised deep learning (SSDL) pipeline employing an attention RNN-based model. Additionally, models like X-Fact (Gupta and Srikumar, 2021) emphasize the integration of evidence with attention architecture. A distinctive approach is taken by FakeFlow (Ghanem et al., 2021), which focus on news articles and study the importance of information flow to detect fake news.

Context plays a pivotal role in understanding and verifying claims. However, its inclusion in datasets is rare, and also varies significantly (Guo et al., 2022). Some datasets, such as those created by Mitra and Gilbert (2015) and Ma et al. (2016), incorporate context derived from related threads. In these instances, a claim is contextualized with a set of pertinent posts, often originating from the same thread, to compensate for the limited context within an individual post. On the other hand, Ghanem et al. (2021) adopt a distinctive strategy, considering the segments of an entire article to analyze information flow. Our work uniquely provides context for sentence-level claims, drawing directly from preceding text or speech. This approach captures the immediate discourse surrounding the claim, highlighting the critical role of textual interactions in credibility assessment. We believe this direct integration of granular context at the sentence level is a novel contribution to the field.

While English remains the predominant language for fake news detection datasets, other languages are often underrepresented. As observed, most efforts to date, such as those by Vlachos and Riedel (2014); Wang (2017), have extracted real-world claims from dedicated English-based websites like Politifact. However, Gupta and Srikumar (2021) are notable exceptions, having curated claims from 25 languages. Their multilingual baselines, including models like Claim Only, Attention-based Evidence Aggregator, and Augmenting Metadata Model, have established a promising foundation for multilingual fact-checking. Another initiative is by Baly et al. (2018), who compiled a dataset of 219 Arabic statements. Both these studies employed Google's evidence retrieval to bolster claim veracity modeling. Yet, their scope appears limited, especially in terms of integrating context and a rich set of metadata/semantic features. No Hebrew-specific dataset for fact verification or fake news detection existed until our contribution, focusing on real-life statements.

**Towards Hebrew Fake-News Detection.** The availability of labeled benchmark datasets is crucial for building statistical approaches for automatic fake news detection in new languages. English has been extensively studied with a large amount of annotated data, while other, less-researched languages, have limited or no annotated data available (Guo et al., 2022).

As of yet, no publicly available Hebrew dataset on fake news detection or claim credibility assessment is publicly available. Furthermore, a limited number of works exist in the field of automatic credibility and falsehood detection in Hebrew. The studies of Dilmon (2004, 2007, 2013) examine the linguistic differences between truthful and deceptive discourse in Hebrew and aim to develop a primary test for the cognitive and emotional processes involved in deception (HaCohen-Kerner et al., 2015).

Although these studies do not analyze natural inputs and instead focus on laboratory-created inputs, they offer valuable insights into the cognitive and emotional processes involved in deception in the Hebrew language. In their corpus, comprising 48 pairs of stories told by 48 subjects, they found several distinguishing features:

- Morphological criteria: False stories exhibit increased use of 3rd person verbs and decreased use of 1st person verbs, while true stories show intensified use of past tense verbs.

- Syntactic criteria: False stories tend to have increased use of dependent clauses and decreased use of independent and conjunction clauses.

- Semantic aspects: False stories involve intensified use of synonym words, relative pronouns, negation, and generalized words.

Leveraging insights from these Hebrew-focused studies, alongside the English-based works, we enhance credebility assessment in Hebrew. Our work extends the existing knowledge base, constructing comprehensive linguistic feature-based models and evaluating their effectiveness on our task.

While our contribution heralds a significant leap in Hebrew-centric datasets for fact verification, it is not confined to that. It further includes a wide range of experiments and the introduction of empirical evaluation of novel models tailored explicitly for this task: Context-Based Model (CBM), Attention-based Evidence Aggregator (Attn-AE) and Context and Evidence Model (CCEM).

## 3 HeTrue: a New Benchmark Dataset for Hebrew Credibility Assessment

In this work, we present HeTrue, a unique and first-of-its-kind Hebrew dataset for credibility assessment, meticulously compiled through a collaboration with professional journalists from "The Whistle", an Israeli fact-checking organization. The dataset comprises 1021 statements from Israeli politicians and public figures, each accompanied by its credibility score.

"The Whistle" operated as an independent NGO from 2017 to 2018 before integrating into Globes newspaper in January 2019. It is noteworthy for being the only Israeli institute complying with the International Fact-Checking Network (IFCN), upholding transparency, impartiality, and fairness in fact-checking. Our partnership with "The Whistle" ensured the integrity of data and alignment with scientific research standards, involving necessary adjustments during collection and annotation.

The statements, spanning one to two sentences,[3] were manually collected by journalists from "The Whistle" in the area between February 2017 to June 2023.

Each statement was assessed on a nuanced 5-point truthfulness scale, ranging from 'True' to 'False', with intermediary labels including 'Mostly-True', 'Partly-True', and 'Mostly-False'. Consistently with recent studies (Gupta and Srikumar, 2021), we also introduced two additional labels: 'Unverifiable' for claims lacking sufficient evidence and 'Other' for cases that do not fit into any of the aforementioned categories. This approach is widely adopted by most fact-checkers (Guo et al., 2022).

To ensure label accuracy, "The Whistle", in partnership with the authors, implemented a rigorous three-stage inter-annotator agreement process. Each statement was initially examined by one professional journalist, followed by an independent review by a second. If disagreements occurred, a third journalist made the final decision, ensuring a highly reliable and consistent dataset. Although this rigorous process resulted in a smaller dataset size than the initially available set of claims, it guarantees a high-quality and reliable dataset.

In addition to the sentence-level statements and their journalist's verified label, we introduce in this dataset the following features: 'claim context' and 'journalist-edited claim'.[4] Crucially, the context encapsulates the sentences around the claim,[5] adding a new dimension to claim assessment. In our work, we examine the improvement gained by incoporating context with the statement itself.

Additionally, we analyzed semantic features including Field, Subject, Title, and tags. This twofold process first involved a "The Whistle" journalist, followed by an editor who reviewed the suggested values and refined them as needed. The full feature list including additional metadata features can be found in Appendix A.6. The label distribution of the dataset is presented in Table 3 in the Appendix.

**Assessment Scenarios and Data Splits**  To gain a better understanding of the underlying task, we created four instances of our dataset for further experimentation.
- FULL-SPECTRUM: The entire dataset, five credibility scores.
- TF-ONLY: Statements only with a "True" or "False" label. Other labels are removed.
- TF-BINS: Following previous studies (Popat et al., 2018; Giachanou et al., 2019), labels are grouped into binary classes, with true, mostly true and half true forming one class (i.e., true), and the rest as false.
- FS-WRITTEN: Statements published in a written media, such as Facebook posts or newsletters. Excluding transcribed statements.

For the experiments described below, we excluded the "Unverifiable" and "Other" claims, leaving five credibility scores for our empirical investigation.

## 4 HeTrue Credibility Assessment

### 4.1 Experimental settings

In this paper, we examine the claim credibility assessment task by establishing a rigorous experimental framework featuring a variety of computational models. Our approach employs both traditional

---

[3]Average statement length (tokens) is 25.

[4]Journalists often revise the original claim for brevity and clarity prior to publication

[5]Average context length is 51 tokens.

linguistic methodologies and advanced deep learning algorithms, each utilizing different aspects of the dataset. This diversified strategy provides us with a broader understanding of the task and its complexities, and sets a comprehensive benchmark for future research. The implemented models are presented in Section 4.3.

## 4.2 Evaluation

In line with previous work (Guo et al., 2022), we employ the macro F1 score as the evaluation metric. To ensure the statistical robustness of our findings, we employ a bootstrap technique (Dror et al., 2018) that involves generating 10,000 resampled variations of the test set. By calculating the standard deviation of the performance metric across these iterations, we evaluate the stability of our results.

**Training Setup** We employed nested cross validation to evaluate the performance of our models. This approach allows us to tune the hyperparameters of the model in an inner loop while evaluating its performance on unseen data in an outer loop. This helps in avoiding overfitting and providing a more robust estimate of model's performance on unseen data. We used k=5 for both inner and outer loop. The hyperparameters are optimized by training the model using different values on the subfolds, and then evaluating the performance of each set of hyperparameters on the sub-fold reserved for testing. The final performance of the model was determined by computing the mean F1 score and standard deviation of the outer loop held-out test set 10,000 times using bootstrap resampling. This evaluation was conducted after optimizing the model through the inner loop. Hence, hyperparameters are optimized independently of the testing data. For hyperparameter optimization we used Optuna (Akiba et al., 2019), a framework for sequential model-based optimization (SMBO) (Hutter et al., 2011) with TPE as the sampling algorithm (Bergstra et al., 2011), to find the best combination of hyperparameter values for a given machine learning model. For further details on preprocessing and hyperparameter tuning, please refer to Appendix A.

## 4.3 Models

**Strong Linguistic Baseline** Drawing from our hypothesis that linguistic cues vary with the credibility of a claim, we engineered linguistic features, based on previous works, from both the statement and its full context, elaborated in Appendix A.

The results, described in Table 1, reveal several noteworthy observations. Firstly, it is clear that the models significantly outperform the metadata models across all four experimental setups. A marked improvement is observed between the full-spectrum and written setups, which is not found in the metadata models. The latter achieved 5% more compared to the former setup. These results align with two key conclusions. First, that there are linguistic cues within the text that can attest to its credibility, which is evident across all four setups and corresponds to the model performance. Second, that these cues are stronger in written statements as opposed to transcribed ones. This finding likely indicates that information is lost or that noise is introduced during the transcription process, such as punctuation marks or writing bias, which impairs the model's learning.

Furthermore, when examining the performance of the model that takes context into account against the model that only considers the statement, it is observed that the former yields superior results (see Table 2). This suggests that additional information present in the context is crucial for prediction. This conclusion aligns with the findings of similar task in news articles (Ghanem et al., 2021). See Appendix A.3 for full feature importance analysis.

**Metadata-based Models** We examine the performance of a model based solely metadata and semantic features, and analyze the combination between these features and the strong linguistic baseline. The Feature list and the preprocessing steps are described in Appendix A.4.

We outline our empirical results in Table 1. We observe that in most dataset setups, the metadata models perform better than their counterparts. In the TF Bins setup, the majority baseline outperforms. We hypothesize that the binning technique might have an adverse impact on the data and feature distribution, affecting the results. Both the metadata feature-based model and the combined model obtain significant improvements and outperform the majority baseline in other setups, indicating that substantial information correlated with claim credibility exists in those features.

Additionally, the combined model achieved better results than the metadata-only model. Analysis of feature importance using Shapley values (SHAPLEY, 1953), shows that the top 5 features in the all-features metadata model are: media source, party, Knesset, gender (objective features), and field (Se-

mantic features). The inclusion of semantic features improves performance significantly.

Finally, we examine the effect of integrating metadata with hand-crafted linguistic features. We found that results were comparable. This was somewhat surprising and it suggests a connection between the linguistic and metadata features. Further research is needed to better understand this relationship. Table 1 summarizes all results including the comparison to a Majority baseline.

## 4.4 Deep Learning Models

Manual extraction of linguistic features might be a demanding task. In addition, such methodology might not take into account hidden patterns in text, and such patterns may be unknown to the researchers or hard to manually extract.

Hence, recent works on credibility assessment and fake news detection exploited various deep neural architectures. In this work, we explore deep learning methods and incorporate recent advancements in the field including state-of-the-art transformers-based contextualized word embeddings in Hebrew (Seker et al., 2021). Additionally, we suggest a novel architecture that combines both evidence and context to enhance the performance on our task.

We conducted experiments in two setups: Statement-Focus and Context-Aware. For each setup we have conducted several experiments as will be described below. We ran our experiments with the Huggingface transformers library Wolf et al. (2020) available under an Apache-2.0 license. For all models, we used the publicly available base checkpoints on Huggingface.[6]

### 4.4.1 Statement-Focus Models

The first category of our models focuses only on the statements. In this setup, we implemented and compared three models.

**Model 1: RNN with Static Word Embeddings**
The first model is based on RNNs using static word embeddings. We experimented with Word2Vec (Mikolov et al., 2013) and fastText (Bojanowski et al., 2017) embeddings, and selected fastText due to its superior performance on our task.

**Model 2: AlephBERT with a Transformer Architecture.** To capture more complex patterns and dependencies in the data, we employed a

contextualized-based model with a transformer architecture. We utilized the state-of-the-art Aleph-BERT (Seker et al., 2021) which was pre-trained on 17.6G of Hebrew text and we fine-tuned it on our task.

**Model 3: Attention-based Evidence Aggregator (Attn-EA)** Building upon the attention-based evidence aggregation (Attn-EA) model proposed by Gupta and Srikumar (2021) – a significant work identified by Guo et al. (2022) – we made refinements to the evidence collection pipeline. Our revised attention-based evidence aggregation model, aggregates evidence from the top five Google search snippets associated with each claim. To reduce potential bias, we curated an 'Excluded Websites' list, which filters out the original source of the claim or any websites that disclose the claim's veracity, such as "The Whistle". Moreover, snippets containing more than 85% of the original claim are omitted. We employed AlephBERT (Augenstein et al., 2019) to separately encode the claim and each piece of evidence, extracting the output of the CLS token: $c, [e_1, e_2, ..., e_n]$. The dot-product attention mechanism (Luong et al., 2015) is then applied to compute attention weights $[\alpha_1, \alpha_2, ..., \alpha_n]$ and a corresponding linear combination: $e = \Sigma_i \alpha_i e_i$. The combined representation is subsequently concatenated with c and fed to the classification layer.

### 4.4.2 Context-Aware Models

The second category of our models incorporates the full context into the analysis. In this setup, we developed and evaluated three models.

**Model 1: RNN with FastText and Full Context**
This model uses the same RNN-based architecture as the statement-focus setup but incorporates the full context into the analysis.

**Model 2: Context-Based Model (CBM)** We propose a model architecture that leverages the power of the AlephBERT model for text understanding and incorporates an attention mechanism to capture the joint interaction between the statement and its context. Given a statement $S$ and its context $C$, we tokenize and encode them using AlephBERT, yielding embeddings $O_S$ and $O_C$. We then employ an attention mechanism, with $O_S$ as the query and $O_C$ as the key and value matrices. The attention mechanism computes the attention weights and generates the merged representation $O_{SC}$. This merged representation captures the con-

---

[6] https://huggingface.co/models

|  | Full Spectrum | TF Only | TF Bins | FS Written |
|---|---|---|---|---|
| Majority | $11.0 \pm 1.2$ | $43.0 \pm 3.0$ | $47.0 \pm 3.3$ | $10.0 \pm 1.1$ |
| Metadata only | $21.1 \pm 1.8$ | $51.6 \pm 3.6$ | $46.9 \pm 3.3$ | $21.9 \pm 1.9$ |
| Metadata+Semantic | $22.5 \pm 1.9$ | $54.7 \pm 3.8$ | $46.6 \pm 3.3$ | $22.6 \pm 2.0$ |
| Linguistic | $25.0 \pm 1.8$ | $60.0 \pm 3.4$ | $51.0 \pm 2.8$ | $30.0 \pm 2.1$ |
| Linguistic+Metadata+Semantic | $25.4 \pm 1.9$ | $60.7 \pm 3.5$ | $50.3 \pm 2.7$ | $31.0 \pm 2.2$ |

Table 1: Comparison of the F1 scores with standard deviations for several models including: Metadata, Metadata+Semantic, Linguistic (Hand-Crafted Features), and Linguistic+Metadata+Semantic. A Majority baseline is also included for comparison. All models were tested on the HeTrue dataset.

|  | Model | Full Spectrum | TF Only | TF Bins | FS Written |
|---|---|---|---|---|---|
| Statement-Focus | Linguistic + M/S | $25.4 \pm 1.9$ | $60.7 \pm 3.5$ | $50.3 \pm 2.7$ | $31.0 \pm 2.2$ |
|  | RNN + fastText | $29.6 \pm 2.3$ | $65.2 \pm 3.4$ | $56.7 \pm 2.7$ | $34.2 \pm 2.3$ |
|  | AlephBert-based | $44.0 \pm 3.0$ | $72.0 \pm 4.1$ | $63.0 \pm 3.5$ | $50.0 \pm 3.0$ |
|  | Attn-EA | $\mathbf{47.8} \pm 3.2$ | $\mathbf{73.5} \pm 4.2$ | $\mathbf{64.9} \pm 3.6$ | $\mathbf{52.7} \pm 3.2$ |
| Context-Aware | Linguistic + M/S | $27.4 \pm 2.2$ | $61.3 \pm 3.3$ | $49.6 \pm 2.5$ | $31.3 \pm 2.2$ |
|  | RNN + fastText | $31.2 \pm 2.4$ | $66.1 \pm 3.5$ | $58.3 \pm 2.8$ | $44.1 \pm 2.4$ |
|  | CBM | $46.0 \pm 3.1$ | $71.0 \pm 4.0$ | $64.0 \pm 3.5$ | $53.0 \pm 3.1$ |
|  | CCEM | $\mathbf{48.3} \pm 3.2$ | $72.8 \pm 4.3$ | $\mathbf{66.7} \pm 3.7$ | $\mathbf{54.5} \pm 3.2$ |

Table 2: The table presents F1 scores for four linguistic models: the hand-crafted feature model, RNN+fastText, AlephBERT, and Evidence-based, in two scenarios - 'Statement only' and 'Full context'. Standard deviations are also reported as subscripts for all F1 scores.

textual interactions between the statement and its context. It is then passed through a fully connected layer with weights $W$ and bias $b$ to produce the output $Z$. Finally, the class probabilities $P$ are obtained by applying the softmax function to $Z$.

**Model 3: Combined Context and Evidence Model (CCEM)** Recent studies showed the importance of evidences and context to claim-assessments tasks. However, models doing so not exist. The Combined Context and Evidence Model (CCEM) architecture leverage the principles of both the Context-Based Model (CBM) and the Attention-based Evidence Aggregator. Initially, similar to the CBM, it forms a merged context-statement representation $O_{SC}$ using the statement $S$ and its context $C$. Concurrently, the model applies the evidence aggregation approach of the Attention-based Evidence Aggregator to obtain a combined evidence representation $E$, derived from the top Google search snippets associated with the claim. The representations $O_{SC}$ and $E$ are then processed through a transformer-based integration layer. This layer uses $O_{SC}$ and $E$ as inputs, conducting a multi-head self-attention operation. The attention mechanism considers $O_{SC}$ as the query and $E$ as the key and value matrices, resulting in

a fused representation $O_{SCE}$ that encapsulates the interactions between the statement, its context, and the external evidence. Finally, the class probabilities $P$ are extracted from $O_{SCE}$ using a softmax function.

### 4.4.3 Results

Table 2 highlights several significant patterns and key findings from our experiments.

One primary observation from our experiments is the consistent outperformance of deep learning models, compared to simpler hand-crafted feature-based models. The latter approaches do highlight some useful features for this task and manage to surpass a number of baseline models, indicating that there are explicit features that can be used for Hebrew credibility assessment task. However, the more complex architectures, including models with static word embeddings and transformer-based models with contextualized embedding, prove to be more effective and deliver better performance overall. A surprising result is the improvement in performance exhibited by the 'Evidence-based' model in the 'Statement only' setup. This model surpasses the performance of the advanced AlephBERT model, even with the CBM architecture, highlighting the significant impact of integrating

external evidence aggregation on model effectiveness.

In the Context-Aware setup, the 'Combined Context and Evidence Model' (CCEM) marginally outperforms other models, highlighting the effectiveness of utilizing both full context and external evidence. The RNN + fastText models also demonstrated substantial improvement when transitioning from 'Statement only' to 'Full context'. This enhancement emphasizes the impact of context in textual understanding, pointing out the limitations of models that focus solely on individual statements.

Interestingly, statement-focused AlephBERT outperforms the Context-Aware RNN model. This could be attributed to AlephBERT's advanced pre-training, which enhances its capabilities in natural language understanding, generalization, and possibly even in encoding writing styles. Throughout the work, we observe that written statements consistently lead in terms of performance. This trend strengthens the hypothesis that there are linguistic and other cues not faithfully captured or altered during text transcription.

## 5 Conclusions

Our study introduces *HeTrue*, a unique and rich dataset designed for automatic credibility assessment in Hebrew. This dataset stands out not only as the first of its kind in Hebrew, but also as the first to incorporate a comprehensive set of features including the statement context, the statement itself, the journalist's version, and additional semantic and metadata features - annotated by professional journalists in a rigours process.

The findings from our experiments underscore the effectiveness of linguistic approaches in the challenging task of credibility assessment. Our results consistently demonstrate enhanced performance when context is integrated into the models across various setups. Furthermore, we show that incorporating evidence-retrieval mechanisms alongside context further contributes to the models' performance.

Despite the advancements made, credibility assessment, particularly in morphologically rich languages like Hebrew, remains a challenging task. While pre-trained language models like AlephBERT show promise in effectively understanding and generalizing language even with limited context, the quest for better accuracy continues. Future research will aim to refine and further improve the

methodologies introduced in this study. Finally, the *HeTrue* dataset, with its unique combination of features, holds potential for a broader range of NLP applications such as stance classification, argument mining, topic modeling, and rumor detection. This broad utility further underscores the value of our contribution and opens new pathways towards more sophisticated fake news detection methodologies.

## 6 Limitations

The dataset is curated by Israeli professional journalists who carefully analyze claims made by public figures and politicians in Hebrew media. However, as stated in the paper, some of the claims are written by the public figures themselves, and others are transcribed by "The Whistle" journalists. This could lead to potential nuances in the way the claims are transcribed, which could influence the results of the analysis. One might assume that the transcription is biased towards the outcome of the claim's truthfulness. To address this issue, we created the written-only setup that evaluates statements that are originally from written media type, such as Facebook, Twitter or opinion articles. Additionally, the dataset used may reflect some collection bias, as the journalists collected the claims that they view as most interesting to the public. This may result in the collection of statements that use provocative language, are false, or have other controversial characteristics. This may limit the generalizability of the findings for naturally occurring texts. Future work could focus on discerning the effects of transcription nuances and overcoming potential biases.

## 7 Ethics Statement

The automatic detection of fake news is a complex task that raises important ethical considerations. One of the key concerns is the risk of using claimants profiling, which could lead to the stigmatization of certain individuals or groups based on their data, such as their gender or political party. Furthermore, there is a risk of bias towards certain ways of expression, thus, linguistic approaches also need to be carefully considered to avoid any form of bias. Additionally, the use of automatic detection systems raises questions about accountability and transparency. The models used in this task can be opaque and difficult to interpret, making it challenging to understand how and why certain decisions are being made. This can make it difficult

to identify and correct errors, and can also make it difficult to hold those responsible accountable for any negative consequences that may arise from the use of these systems. These concerns are amplified in today's information landscape where data can be created or manipulated by Generative Artificial Intelligence, further complicating the discernment of facts from fiction and truths from lies. Given these ethical considerations, we urge careful handling of automatic credibility assessment and encourage inclusive, responsible system development.

## 8 Broader Impact

This paper is submitted in the wake of a tragic terrorist attack perpetrated by Hamas, which has left our nation profoundly devastated.

On October 7, 2023, thousands of Palestinian terrorists infiltrated the Israeli border, launching a brutal assault on 22 Israeli villages. They methodically moved from home to home brutally torturing and murdering more than a thousand innocent lives, spanning from infants to the elderly. In addition to this horrifying loss of life, hundreds of civilians were abducted and taken to Gaza. The families of these abductees have been left in agonizing uncertainty, as no information, not even the status of their loved ones, has been disclosed by Hamas. This is *not* fake new. This is first-hand evidence from our family, friends, relatives and acquaintances, the ones that survived the horror.

The heinous acts committed during this attack, which include acts such as shootings, sexual assaults, burnings, and beheadings, are beyond any justification.

Beyond the loss we suffered as a nation and as human beings due to this violence, many of us feel abandoned and betrayed by members of our research community who did not reach out and were even reluctant to publicly acknowledge the inhumanity and total immorality of these acts.

This horrific situation highlights the devastating impact of misinformation, fake news and propaganda, and what they can inflict on society. In this case, manipulated stories and misleading narratives contributed to justifying or diminishing the severity of Hamas' atrocities, underscoring the urgent need for improved methods in assessing the credibility of information, such as the methods advocated for and developed in this work.

We fervently call for the immediate release of all those who have been taken hostage and urge the academic community to unite in condemnation of these unspeakable atrocities committed by Hamas, who claim to be acting in the name of the Palestinian people. We call all to join us in advocating for the prompt and safe return of the abductees, as we stand together in the pursuit of justice and peace.

This paper was finalized in the wake of these events, under great stress while we grieve and mourn. It may not be as polished as we would like it to be.

## 9 Acknowledgments

We would like to thank "The Whistle" at "Globes" for their essential support in creating the HeTrue dataset, serving as a foundational element of our paper and facilitating future research in this domain. This research was funded by the Israeli Ministry of Science and Technology (MOST) grant No. 3-17992, and an Israeli Innovation Authority grant (IIA) KAMIN grant, for which we are grateful. In addition, This project has received funding from the European Research Council (ERC) under the European Union's Horizon 2020 research and innovation programme, grant agreement No. 677352.

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

# A Appendix

## A.1 Text preprocessing

Before creating and extracting the linguistic features, we prepare our data for that. Cleansing(cleaning) the dataset to correct corrupt or inaccurate records and extracting basic linguistics elements that will be used later to create the features. To do so , we used YAP (Yet Another Parser)More et al., 2019 developed by Reut Tsarfaty and Amir More. To preprocess the statements, the following stages were taken:

- Removing irrelevant special characters
- Removing double qoutes for acronyms
- Removing Stop words (but save the original text separately)
- Tokenization - for specific features that are token-based.
- Stammer - for specific features that are stem-bases.
- Lemmatizer - for specific features that are lemma-based.

In addition we used YAP to extract the following information to later use for feature extraction:

- POS tagging
- Dependencies parts extractions
- Grammatical person
- Verbs tense

## A.2 Linguistic features extraction

We developed a set of hand-crafted linguistic features based on previous studies that have indicated a correlation between these features and the psychological state of claimants. This psychological state can be used to infer the truthfulness of a claimant statement, as it is reflected in the language of the claim. The features are divided into the following categories:

- Lexical features – word based
  - Bow tf-idf
  - Complexity measure
  - Readability measure
  - Average word length
  - Count of H' hayedia
  - Count of distinct words
  - Count of words
  - Count of syllables
  - Usage of uncertainty words
  - First names usage
  - Last names usage
- Lexical features – char based
  - Total number of chars
  - Digits percentage
  - Letters percentage
  - Bag of chars tf-idf
- Syntactic features
  - Count of punctuation marks
  - Count of special chars including @$%^<>
  - Count per dependency parts : dependencies between words (sub-obj for instance)
  - Count per POS
  - Count per Person(first,second,third) in a statement.
  - Count per verb tense (BEINONI, PAST, FUTURE, IMPERATIVE).
- Structural features
  - Number of inner quoutes
  - Hashtags count
- Semantic features
  - Positive, negative, and objective semantic based on Bert based model built on tagged tweets (Twitter).

We extracted these features on both claim and context.

## A.3 Strong Linguistic Baseline - Features Analysis

We conducted a feature importance analysis using Shapley values on our best XGBoost model after optimization using the nested cross validation approach. The key finding are as follows. First, false statements tend to make greater use of relative clauses and punctuation marks, particularly commas. Second, true statements tend to have a more positive sentiment and make greater use of complement clauses and inseparable prepositions. The use of definite markers is more prevalent in false statements. Next, true statements tend to make use of words with greater complexity as measured by the Flesch-Kincaid index (Flesch, 1965). In addition, the overall statement length was not found to be a significant factor in determining truthfulness.

## A.4 Metadata features

The dataset we curated from "The Whistle" contains in addition to the claim additional metadata features. These features can be divided to semantic

features and objective features. The objective features are: date, speaker, role, party, media, knesset and gender. The semantic features are: subject, title, field, and tags.

**Metadata preprocessing** One of the "metadata features" is 'media'. The 'media' feature describes the platform in which the claim was produced. The way "The Whistle" journalists described the platform is in the most granular level. For example, a claim produced in a radio station called "GLTZ" in a radio program called "Nahon Le-Haboker", was described in the 'media' feature as "Nahon Le-Haboker GLTZ". We hypothesis that adding the type of media, the media source and the producer type(spoken/written) will improve the accuracy of our predictions. This is because different types of media can have different effects on people. For example, a video on YouTube may have a different effect than a news article on a website. Additionally, different media sources can have different biases, which may influence how an individual interprets and reacts to a piece of media.

## A.5 Hyperparameter Optimization

As stated in the training setup section, we employed nested cross validation to evaluate the performance of our models and to choose the best hyperparameters. For the hand-crafted feature model , the metadata models and the integrated (metadata + hand-crafted features) models. We used XGBoost as the classification model and we searched among the following parameters: we searched `booster` in dart ,gbtree ,gblinear and chose gbtree. We searched `lambda` in 1e-8, 1.0 and chose 6.421 . We searched `alpha` in 1e-8, 1.0 and chose 0.0008 . We searched `min_child_weight` in 1-10 and chose 6. We searched `subsample` in 0.01 - 1.0 and chose 0.385. We searched `colsample_bytree` in 0.01 - 1.0 and chose 0.4487. We searched `max_depth` in [1-9] and chose 5. We searched `eta` in [1e-8 - 1.0] and chose 0.7542. We searched `gamma` in [1e-8 - 1.0] and chose 1.005 . We searched `grow_policy` in "depthwise", "lossguide" and chose lossguide. Also we implemented feature selection using sklearn SelectKBest method. We search `k` in [100-8000] and chose 2127. Also, we examine the use of up-sampling method to overcome the unbalance of our data. We chose between upsampling the True score statements, the True score statements + Mostly true statements or upsampling all to count

| Label | # Claims |
|---|---|
| True | 86 |
| Mostly true | 102 |
| Half true | 122 |
| Mostly False | 252 |
| False | 343 |
| Unverifiable | 88 |
| Other | 28 |

Table 3: Label distribution in HeTrue dataset

as the majority class. We finally chose to upsample only the True score statements. We chose if to use the original statement with or without stopwords, and chose without. For the full context as input we chose with stopwords(original) as it performed the best.

For the Deep learning models, we experiment with two variations: classification approach using softmax in the final layer and ordinal regression approach using the method that was introduced by (Cheng et al., 2008) leveraging the ordinal characteristic of our labels.

For the RNN model we examined both word2vec and fastText and we chose fastText. Also, we compare the performance of GRU vs LSTM and we chose GRU. The architecture chose is: Dropout 0.5 after the embedding layer. than GRU layer with dim=8. Than dropout of 0.2 and sigmoid as activation function for the final layer (for ordinal model) and softmax (for classification model). The batch size chosen is 32 and number of epochs 10. We also used the same upsampling method desribed above. Here, we also considered the use of stopwords, and the decision was the same as above.

For the AlephBERT model we searched the learning rate in 1e-6, 5e-5, 1e-5 and chose with 1e-4. We experiment two architectures one with two fully connected layers on top of pooled output of the pre-train AlephBERT and the other is one fully connected layer. Both with dropout layer and ReLU as activation function. We finally chose : dropout 0.5 (searched in [0-0.5] , and one fully connected layer.We searched the batch size in 16, 32, 64 and chose 8. To optimize performance we used gradient accumulation for a total batch size of 16. We searched weight decay in 0, 0.1 and chose 1e-4. We also used the same upsampling method desribed above. Here, we also considered the use of stopwords, and the decision was the same as above.

## A.6 HeTrue Dataset Features and Definitions

Dataset label distribution can be found in table table 3 Feature list of HeTrue dataset described below.

**Journalist-edited claim** The claim as appeared on "The Whistle" website. Sometimes the claim is being transformed by the website editors for various reasons.

**Original claim** The claim as produced by the claimant and appeared in the original media.

**Claim Full context** The original claim and the surrounding sentences, up to two sentences before and after the claim.

**Claim date** The claim's producing date.

**Claimant** The person full name who made the claim.

**Gender** Male/Female.

**Role** The claimant's role.

**Party** The claimant's political party.

**Field** General subject area or discipline that the claim relates to.

**Subject** Specific topic that the claim is about.

**Title** A brief, descriptive phrase that summarizes the main idea of the claim.

**Media** The media in which a claim is produced.

**Knesset#** The Knesset number in which the claimant is part of.

**Tags** Added by "The Whistle" journalists to label and categorize the claims.

**Label** "Truth score" given by the "The Whistle" journalists.

## A.7 Comparison of Credibility Assessment Datasets

See Table 4 for a comprehensive comparison of credibility assessment datasets, focusing on sentence-level claims. For each dataset, we present the number of instances, the language, the number of labels and the source of the dataset. Additionally, we highlight the comprehensiveness of each dataset in terms of additional features: semantic annotation, metadata features, and context. Semantic annotations pertain to features extracted by professional journalists. Metadata features cover attributes related to the claim and claimant. Context encompasses surrounding text relevant to the statement. *HeTrue* stands out as the sole dataset integrating all three of these additional features.

## A.8 Dataset Example

An example from the HeTrue dataset can be found in Table 5 below.

| Dataset | #Inputs | Semantic Annot. | Metadata Feat. | Context | Classes | Sources | Lang |
|---|---|---|---|---|---|---|---|
| CrimeVeri (Bachenko et al., 2008) | 275 | N/A | N/A | N/A | 2 | Crime | En |
| Politifact (Vlachos and Riedel, 2014) | 106 | N/A | Yes | N/A | 5 | Fact Check | En |
| StatsProperties (Vlachos and Riedel, 2015) | 7,092 | N/A | N/A | N/A | Numeric | Internet | En |
| Emergent (Ferreira and Vlachos, 2016) | 300 | N/A | N/A | N/A | 3 | Emergent | En |
| CreditAssess (Popat et al., 2016) | 5,013 | N/A | N/A | N/A | 2 | Fact Check/Wiki | En |
| PunditFact (Rashkin et al., 2017) | 4,361 | N/A | N/A | N/A | 2/6 | Fact Check | En |
| Liar (Wang, 2017) | 12,836 | N/A | Yes | N/A | 6 | Fact Check | En |
| Verify (Baly et al., 2018) | 422 | N/A | N/A | N/A | 2 | Fact Check | Ar/En |
| CheckThat18-T2 (Barron-Cede´no et al., 2018) | 150 | N/A | N/A | N/A | 3 | Transcript | En |
| Snopes (Hanselowski et al., 2019) | 6,422 | N/A | Yes | N/A | 3 | Fact Check | En |
| MultiFC (Augenstein et al., 2019) | 36,534 | N/A | Yes | Yes | 2–27 | Fact Check | En |
| Climate-FEVER (Diggelmann et al., 2020) | 1,535 | N/A | N/A | N/A | 4 | Climate | En |
| SciFact (Wadden et al., 2020) | 1,409 | N/A | N/A | N/A | 3 | Science | En |
| PUBHEALTH (Kotonya and Toni, 2020b) | 11,832 | N/A | Yes | N/A | 4 | Fact Check | En |
| COVID-Fact (Saakyan et al., 2021) | 4,086 | N/A | N/A | N/A | 2 | Forum | En |
| X-Fact (Gupta and Srikumar, 2021) | 31,189 | N/A | Yes | N/A | 7 | Fact Check | Many |
| **HeTrue** | **1021** | **Yes** | **Yes** | **Yes** | **7** | **Fact Check** | **He** |

Table 4: Overview of datasets for credibility assessment, focused on individual statements.

| | |
|---|---|
| Edited claim | בשנים האחרונות ועדות השחרורים נתנו לפחות ופחות אנשים שליש, רק כ-25% מהאנשים קיבלו שליש. 

 In recent years, the parole boards have given fewer and fewer people a third, only about 25% of the people received a third. |
| Original claim | צריך לדעת שבשנים האחרונות ועדות השחרורים נתנו לפחות ופחות אנשים שליש, רק כ-25% מהאנשים קיבלו שליש, וכחלק מהפתרון שלנו, בג"ץ הרי אמר שצריך להגדיל את תנאי המחייה של האסירים, ולכן צריך גם, בין השאר, לצמצם את כמות האסירים. 

 You should know that in recent years the parole boards have given fewer and fewer people a third, only about 25% of the people received a third, and as part of our solution, the High Court of Justice said that the living conditions of the prisoners should be increased, and therefore the number of prisoners should also, among other things, be reduced. |
| Full context | אני אמרתי גם למשרד המשפטים וגם בתקשורת שכשוועדת שחרורים נותנת שליש, הקריטריונים צריכים להיות התנהגות טובה ומסוכנות, ולא צריך להמציא כל מיני קריטריונים אחרים שלא נמצאים בחוק, ולכן כל אסיר שמגיע בפני ועדת שחרורים, דעתי היא שאם התנהגותו הייתה טובה ואם הוא לא מסוכן אז צריך לתת לו שליש, ועל אחת כמה וכמה אם הוועדה נותנת לו שליש, אני לא חושבת שצריך לערער, אלא במקרים מאוד קיצוניים. צריך לדעת שבשנים האחרונות ועדות השחרורים נתנו לפחות ופחות אנשים שליש, רק כ-25% מהאנשים קיבלו שליש, וכחלק מהפתרון שלנו, בג"ץ הרי אמר שצריך להגדיל את תנאי המחייה של האסירים, ולכן צריך גם, בין השאר, לצמצם את כמות האסירים. 

 I told both the Ministry of Justice and the media that when the parole board gives a third, the criteria should be good and dangerous behaviour, and there is no need to invent all kinds of other criteria that are not in the law, so every prisoner who comes before a parole board, my opinion is that if his behaviour was good and if he is not dangerous, then he should To give him a third, and even more so if the committee gives him a third, I don't think it is necessary to appeal, except in very extreme cases. You should know that in recent years the parole boards have given fewer and fewer people a third, only about 25% of the people received a third, and as part of our solution, the High Court of Justice said that the living conditions of the prisoners should be increased, and therefore the number of prisoners should also, among other things, be reduced. |
| Claim date | 06.05.2018 |
| Claimant | Ayelet Shaked |
| Gender | Female |
| Role | Minister of Justice |
| Party | Yamina |
| Field | Law and Order / חוק ומשפט |
| Subject | Release of prisoners / שחרור אסירים |
| Title | On the approval of early releases for prisoners / על אישור שחרורים מוקדמים לאסירים |
| Media | Kalman-Lieberman - Kan-Bet / קלמן-ליברמן - כאן ב' |
| Knesset# | The 20th Knessett / הכנסת ה-20 |
| Tags | Ayelet Shaked, prisons / איילת שקד,בתי סוהר |
| Label | True |

Table 5: Example from HeTrue dataset. For reference, translations are also shown