# OpenReview forum: "The Truth, The Whole Truth, and Nothing but the Truth:  A New Benchmark Dataset for Hebrew Text Credibility Assessment"
_EMNLP/2023/Conference — EMNLP 2023 Findings_

### Official Review · Reviewer_BPdV · 2023-08-03

**Soundness:** 3

**Excitement:**

3: Ambivalent: It has merits (e.g., it reports state-of-the-art results, the idea is nice), but there are key weaknesses (e.g., it describes incremental work), and it can significantly benefit from another round of revision. However, I won't object to accepting it if my co-reviewers champion it.

**Paper Topic And Main Contributions:**

The paper introduces HeTrue: a new, publicly available dataset for evaluating the credibility of statements made by Israeli public figures and politicians. The dataset consists of 1021 statements, manually annotated by Israeli professional journalists, for their credibility status.
Experiments are conducted to establish a baseline and compare text-only methods with others using additional data like metadata, context, and evidence. Several credibility assessment models are developed including a feature-based model that utilizes linguistic features, and state-of-the-art transformer-based models with contextualized embeddings from a pre-trained encoder. Results from the experiment show that:
•	the best performance is achieved when models integrate statement and context
•	the HeTrue is a challenging benchmark and can be used in the future for model training and evaluation.


**Questions For The Authors:**

How the findings in your experiments about the relative performance of different models and approaches are compared to the finding from experiments with other datasets and languages?



**Reasons To Accept:**

•	A well-structured and well-written paper, including the attachments.
•	The area of automatic claim verification is a hot topic.
•	A new dataset is collected and annotated by human experts.
•	The collection and annotation of the dataset is done very professionally, with care to all details.
•	Experiments are conducted to establish a baseline and state-of-the-art approaches are applied.
•	The conducted experiments are well-designed.
•	The new dataset is suitable as a benchmark.


**Reasons To Reject:**

Limited discussion of how the findings of the experiments are compared with other works. Do the models perform in the same/similar/different way?

**Reproducibility:**

4: Could mostly reproduce the results, but there may be some variation because of sample variance or minor variations in their interpretation of the protocol or method.

**Reviewer Confidence:**

4: Quite sure. I tried to check the important points carefully. It's unlikely, though conceivable, that I missed something that should affect my ratings.

**Typos Grammar Style And Presentation Improvements:**

The first part of the title “The Truth, The Whole Truth, and Nothing but the Truth:” can be obliterated.
And some of the motivation in the introduction could be shortened. This will give a bit of space to move some parts from the appendixes to the main body of the paper.

---

> ### Author Rebuttal · Authors · 2023-08-29
>
> We would like to extend our sincere gratitude to the reviewers for their meticulous evaluation and constructive feedback. We deeply value the insights provided, which have offered us the opportunity to enhance the quality and clarity of our work. In this response, we have systematically addressed each of the concerns raised and have made corresponding revisions to our manuscript where necessary.
>
>
>
> **Response to the Reviewer's Query on Model Comparisons Across Datasets and Languages**:
>
> We appreciate your question regarding the relative performance of our models and how they compare with experiments on other datasets and languages.
>
> In our experiments, all models incorporating context consistently outperformed their counterpart models that did not use context. As far as we know, we are the only to establish experiments focusing on sentence-level claims with their context. However, different context types of different claim types examined before. Our finding is congruent with existing literature where the integration of context has demonstrably enhanced model performance. For instance:
>
> 1. **Ghanem (2021)** in "Fakeflow: Fake News Detection by Modeling the Flow of Affective Information'' showcased the value of leveraging context to bolster fake news detection. They segment news articles, and treat each segment as “claim” and the other as context.
>
> 2. **Nakashole and Mitchell (2014)**, "Language-aware Truth Assessment of Fact Candidates", proposed a methodology that harnesses Subject-Predicate-Object (SPO) triplets extracted from the Web, also can be considered as context,  emphasizing language objectivity analysis to determine veracity.
>
> 3. **Mitra and Gilbert (2015), Ma et al. (2016), and Zhang et al. (2021)** leveraged posts related to the main post (which needed verification) as context. This approach resulted in improved model performance.
>
>
> In our work, we closely observed how different evidence forms influenced model performance. As highlighted in the literature, evidence plays a pivotal role, especially in the realm of automated fact-checking. However, the nature and structure of this evidence can often dictate the efficacy of the model.
>
>
> 1. **Popat et al. (2018)**, through "DeClarE: Debunking Fake News and False Claims using Evidence-Aware Deep Learning", accentuated the utility of external evidence. Their work closely mirrors our findings, where models leveraging external sources, especially textual ones, generally showcase heightened performance accuracy.
>
> 2. **Guo (2022)** in "A Survey on Automated Fact-Checking" conducted a comprehensive dive into various evidence types used for fact-checking. His study corroborated our observation that textual sources, spanning from news articles and academic papers to Wikipedia documents, remain among the most dominant and effective forms of evidence in the space.
>
> However, while textual sources are prevalent and widely accepted, they are not without challenges. As highlighted in the cited literature, using metadata as evidence might offer complementary information but lacks the depth to ground claims conclusively (Wang, 2017; Potthast et al., 2018). Similarly, though structured knowledge, such as knowledge bases, presents an organized approach, their inherent incompleteness can sometimes impede conclusive evidence generation (Bordes et al., 2013; Socher et al., 2013).
>
> In our experiments, we found a semblance of these trends and challenges. Our models that embraced richer textual evidence and blended it with metadata / semantic features demonstrated a nuanced ability to tackle fake news detection, resonating with findings from works like Vlachos and Riedel (2015) and Gupta et al. (2020).
>
> Our findings on evidence-based models align with the broader trends and insights of the NLP community.
>
>
> Recent research underscores the significance of either context, evidence, or both, in various capacities. While our findings resonate with these works, a distinguishing feature of our approach is the simultaneous integration of both context and evidence, an amalgamation not extensively explored in the literature to date. The novel approach that we suggested outperforms past architecture, and we believe it can encourage the community to further research this avenue.
>
> *We're grateful for you bringing to our attention the typos and grammatical mistakes; they have been corrected in the final version of the manuscript.
>
> *All references cited in this response can be found in the bibliography section of our paper.
>
> We deeply appreciate the time and dedication the reviewer has committed to evaluating our manuscript. We have endeavored to address each concern with thoroughness and rigor. It is our sincere hope that, upon reviewing our revisions and the effort we have put into addressing the feedback, the reviewer finds our submission to be significantly enhanced and more robust. Thank you once again for your invaluable insights.

---

### Official Review · Reviewer_UQD5 · 2023-08-04

**Soundness:** 1

**Excitement:**

2: Mediocre: This paper makes marginal contributions (vs non-contemporaneous work), so I would rather not see it in the conference.

**Paper Topic And Main Contributions:**

This paper presents a dataset containing statements from Israeli public figures and politicians. The dataset is annotated by professional journalists and different techniques were used for evaluating the dataset. The proposed model achieves a 48 F1 Score as the best score.



**Reasons To Accept:**

The dataset can be used to develop more robust models.

**Reasons To Reject:**

This paper presents a dataset containing statements from Israeli public figures and politicians. The dataset is annotated by professional journalists and different techniques were used for evaluating the dataset. The proposed model achieves a 48 F1 Score as the best score. However, how the bias was removed in the annotation process. Moreover, it is very unclear how the proposed dataset can be reproduced or adds new knowledge in terms of advancing the field.



**Reproducibility:**

1: Could not reproduce the results here no matter how hard they tried.

**Reviewer Confidence:**

5: Positive that my evaluation is correct. I read the paper very carefully and I am very familiar with related work.

---

> ### Author Rebuttal · Authors · 2023-08-29
>
> We would like to extend our sincere gratitude to the reviewers for their meticulous evaluation and constructive feedback. We deeply value the insights provided, which have offered us the opportunity to enhance the quality and clarity of our work. In this response, we have systematically addressed each of the concerns raised and have made corresponding revisions to our manuscript where necessary.
>
>
> **Response to reviewer Bias concern:**
>
>
> We apologize for not highlighting this crucial aspect sufficiently in our initial submission. We understand the paramount importance of clarifying any potential biases, especially in datasets like ours, and are committed to elucidating this further in our final version. Bias Mitigation in Annotation:
>
> * **Multi-level Review:** As outlined in our paper, we used a rigorous three-stage inter-annotator agreement procedure. Each statement underwent initial examination by one journalist, then an independent review by a second journalist. Any discrepancies between their assessments brought in a third journalist for the final decision. This minimized potential bias from any individual journalist.
>
> * **Diverse Annotation Team:** Journalists annotating the data hail from diverse backgrounds and maintain "The Whistle's" professional standards. Their commitment to objectivity ensures labels are based on factual accuracy, free from personal or organizational biases.
>
> * **IFCN Guidelines:** The participating journalists strictly follow the IFCN guidelines, which uphold transparency, impartiality, and fairness in fact-checking.
>
> * **Nuanced Labeling:** We introduced a detailed 5-point truthfulness scale, complemented by 'Unverifiable' and 'Other' categories. This granularity captures the complexities of real-world statements, avoiding potential bias and oversimplification.
>
> * **Comparison with Established Methods:** Our labeling process aligns with methods employed in recent fact-checking studies, highlighting its consistency with prevailing community standards.
>
> * **Transparent Scoring Rationale:** To further reduce bias and ensure transparency, the entire rationale behind our scoring system is publicly available. Additionally, for each claim annotated, the specific rationale behind its score is clearly documented and made transparent. This level of openness allows for external scrutiny, assuring users and researchers of the thoroughness, objectivity, and integrity of our scoring methodology.
>
> Though our rigorous process yielded a smaller dataset, it ensures the highest quality and reliability.
> Lastly, all text-based features were double-checked for accurate transcription. The semantic features benefited from a two-tier verification process involving both a "The Whistle" journalist and an editor. We trust this elucidation sufficiently addresses your concerns.
>
>
>
> ** Response to reviewer Contribution concern**
>
> We deeply value your feedback and understand the significance of clearly articulating the distinctiveness of our research. We believe our work embodies contributions in the NLP domain, and we're committed to elucidating these in detail.
>
>
> * **Introduction of "HeTrue" Dataset:** A principal contribution is the development of the "HeTrue" dataset. Distinctively, this dataset captures both claims and their contextual surroundings, establishing a potential reference point for subsequent datasets in this field. The dataset is curated in a thorough process with unique features (such as semantic features, edited claim and context).  As far as we know, this is the first-ever claim assessment dataset that accompanies each claim with its context, in any language.  More details can be found in table 4.
>
> ** Extensive Experimental Analysis:** A cornerstone of our research is the comprehensive experimental framework we employed. We systematically investigated various modeling types and techniques, encompassing both traditional NLP methods and cutting-edge transformer architectures and designed unique transformer based models. We believe that such a broad and deep experimental setting provides the community with a clearer understanding of the strengths and limitations of different techniques when applied to the challenge of credibility assessment.
>
> * **Linguistic Analysis in Hebrew:** Our work delves into the linguistic nuances of Hebrew, addressing a gap in datasets oriented towards fake news detection. We identified particular linguistic markers in Hebrew that distinguish between credible and non-credible statements (Appendix A3). This contributes to our understanding of linguistic variations in credibility assessment, with possible implications for other languages. Our hand-crafted linguistic features, informed by previous studies, spotlight essential cues in credibility detection.
>
> * **Integration of Context:** We've highlighted the salience of statement context in the analysis. Our work shows that the combination of statements with their contexts can significantly enhance predictive accuracy, a finding that has implications for both further research and practical applications. As previously noted, there are no other works that focus on sentence-level claims (not from news articles) paired with context. We hope that our findings will stimulate the creation of such datasets in the future.
>
> * **Interplay of Metadata and Linguistic Features:** Our experiments integrating metadata with linguistic attributes presents intriguing findings. As detailed in Table 1, our results suggest a nuanced relationship between these two domains.
>
>
>
> * **Attention-based Evidence Aggregator (Attn-EA) Model:** Expanding on the work by \citet{gupta2021x}, our Attn-EA model offers a refined approach to evidence curation, ensuring a meticulous integration of evidence from top Google search snippets and rigorous bias filtering.
>
> * **Advancement in Context-Aware Models:** We introduced models such as the Context-Based Model (CBM) and the Combined Context and Evidence Model (CCEM). The CCEM, in particular, integrates features from both CBM and Attn-EA, providing a more rounded approach by integrating statements, their contexts, and external evidence. Our approach uniquely combines statement context with external evidence, addressing an aspect not frequently explored in prior research. The methodologies we employ, particularly in models like CCEM, contribute a novel perspective to credibility evaluation.
>
>
> In summary, we believe our work offers novel insights and methodologies in several areas of NLP research. We are grateful for the opportunity to elucidate our contributions more comprehensively.
> In light of our clarifications, we sincerely hope that the distinctiveness and value of our contributions become apparent.
>
> ---
>
>
>
>
>
>
>
> ** Response to reviewer reproducibility concern**
>
> We sincerely appreciate your feedback regarding reproducibility concerns. The integrity and transparency of our research is of utmost importance, and we understand the value of ensuring our results can be easily replicated by the community. To address the concern you raised:
>
> It is important to emphasise that both the dataset and the codebase will be made publicly available upon acceptance of the paper.
>
> For all reported experimental results:
>
> We have evaluated all our models across four dataset setups, detailed in Section 3. The experimental setup, encompassing both evaluation and training, is elaborated upon in Section 4.
>
> **Strong Linguistic Baseline:** Various models were examined, with Xgboost being selected based on hyperparameter optimization, as delineated in Appendix A.5. The feature extraction process involves text pre-processing steps, exhaustively described in Appendix A.1 and A.2. We extracted features from both the statement and its encompassing context.
>
> **Metadata-based Models:** Several models based on metadata were utilized. All feature extraction processes, inclusive of pre-processing, are meticulously detailed in Appendix A.4.
>
> **Deep Learning Models:** We have approached experiments in two distinct setups: Statement-Focus and Context-Aware. For each:
>
> - **Statement-Focus Models:** Three models were employed. This included the RNN with Static Word Embeddings utilizing fastText embeddings, selected for its exemplary performance post hyperparameter optimization (detailed in the appendix). Next is the AlephBERT model, a Hebrew BERT variant fortified with a Transformer architecture. It underwent pre-training on 17.6G of Hebrew text and fine-tuning for our specific task. The Attention-based Evidence Aggregator (Attn-EA) model leverages AlephBERT for encoding both claims and evidence, utilizing dot-product attention, as outlined in Section 4.4 (Full formula can be found there too).
>
> - **Context-Aware Models:** Our experiments encompassed the RNN with FastText and Full Context. The Context-Based Model (CBM) and the Combined Context and Evidence Model (CCEM) both employ AlephBERT. The latter integrates statement, context, and external evidence from Google search snippets for assessment. Comprehensive formulas are available in Section 4.4. Complete parameter settings are listed in Appendix A.5.
>
> **Source code** will be made publicly accessible upon acceptance, accompanied by precise specifications and dependencies.
>
> **Evaluation Measure/Statistics:** Metrics, including the macro F1 score, are elucidated in tables 1 and 2 of Section 4. We employ the bootstrap technique to ensure statistical robustness. Comprehensive details are presented in Section 4.2.
>
>
>
> **Regarding the dataset:**
>
> - **Dataset Statistics:** HeTrue encompasses 1021 statements with label distribution outlined in Table 3.
> - **Train/Validation/Test Splits:** We leveraged a nested cross-validation strategy with 5-folds in both inner and outer loops. The detailed approach and methodologies are documented in the appendix and Section 4.
> - **Data Exclusions and Pre-processing:** Only the labels "Unverifiable" and "Other" were omitted. All remaining pre-processing phases are exhaustively detailed in Section 3 and the appendix.
> - **Language:** The dataset is entirely in Hebrew.
> - **Dataset Link:** This will be shared upon paper acceptance.
> - **Data Collection Process:** Detailed in Section 3, including our collaboration with "The Whistle" and the three-stage inter-annotator agreement process.
>
> For all models, we used the publicly available base checkpoints on Huggingface, with Pytorch 2.0.
> All hyperparameters details for our models, described thoroughly in the appendix A.5. We have used Optuna library.
>
>
>
> We hope this comprehensive response should adequately address reproducibility concerns.
>
> –
> *All references cited in this response can be found in the bibliography section of our paper.
>
> We deeply appreciate the time and dedication the reviewer has committed to evaluating our manuscript. We have endeavored to address each concern with thoroughness and rigor. It is our sincere hope that, upon reviewing our revisions and the effort we have put into addressing the feedback, the reviewer finds our submission to be significantly enhanced and more robust. Thank you once again for your invaluable insights

---

### Official Review · Reviewer_uzQB · 2023-08-05

**Soundness:** 3

**Excitement:**

3: Ambivalent: It has merits (e.g., it reports state-of-the-art results, the idea is nice), but there are key weaknesses (e.g., it describes incremental work), and it can significantly benefit from another round of revision. However, I won't object to accepting it if my co-reviewers champion it.

**Missing References:**

I would have liked a more focused related work section also describing similar dataset for other non-English / underrepresented languages. Also, the context and evidence based methods are very shallowly presented in the related work.

**Paper Topic And Main Contributions:**

The paper propose a dataset, called HeTrue, for text credibility assignment / fact checking in Hebrew. The dataset contains information collected from the Whistle website [1], including some metadata about the source of the statement being investigated and some textual evidence about the credibility assignment decision.

The authors also use HeTrue for a series of experiments, both using feature engineering and standard ML and using AlephBERT-based solutions. The main contribution with regard to experiments is the usage of context and evidence together with the statement under consideration using some additional attention mechanisms (which are different for Attn-EA, CBM, CCEM models).

[1] - https://thewhistle.globes.co.il/feed

**Questions For The Authors:**

A. Not clear how the data has been gathered from The Whistle - is it just scraped / is the dataset released together with the Whistle?

B. I would have liked a more focused related work section also describing similar dataset for other non-English / underrepresented languages. Also, the context and evidence based methods are very shallowly presented in the related work. Can you tackle this points in the final version? From my side, you can remove some more generic parts about using network analysis for fact checking as they are completely unrelated to the current work.

C. I would have liked to see some simpler baselines for using context and evidence, e.g. just concatenating the two with a <SEP> token or another special token. Have you tried any of these simpler experiments? This would justify that the proposed Attn-EA, CBM, CCEM models add value compared to a simpler method.

D. Can you provide an example of statement, evidence and context in the Appendix?

E. How do you know that the context and evidence do not contain any information about the actual credibility label for the statement? Are these manually inspected?

F. How where the statements in the HeTrue dataset selected?

**Reasons To Accept:**

* New dataset for text credibility assignment / fact checking in Hebrew. Dataset quality looks very good, and also provides information like metadata, context and evidence.
* Good explanation for the experiments, nice integration of context and evidence in the final decision.

**Reasons To Reject:**

* Not clear how the data has been gathered from The Whistle - is it just scraped / is the dataset released together with the Whistle?
* I would have liked a more focused related work section also describing similar dataset for other non-English / underrepresented languages. Also, the context and evidence based methods are very shallowly presented in the related work.
* I often felt that some sections are longer and without added value to the paper - maybe a short paper format might have been better.
* I would have liked to see some simpler baselines for using context and evidence, e.g. just concatenating the two with a <SEP> token or another special token .

**Reproducibility:**

3: Could reproduce the results with some difficulty. The settings of parameters are underspecified or subjectively determined; the training/evaluation data are not widely available.

**Reviewer Confidence:**

3: Pretty sure, but there's a chance I missed something. Although I have a good feel for this area in general, I did not carefully check the paper's details, e.g., the math, experimental design, or novelty.

**Typos Grammar Style And Presentation Improvements:**

The paper might use an extra round of proofreading, e.g.

- The former argue that => argues
- This study use both => uses
- For example Long et al. (2017) show => For example, Long ...
- English has been focused on with much more annotated ... => rephrase

---

> ### Author Rebuttal · Authors · 2023-08-29
>
> We would like to extend our sincere gratitude to the reviewers for their meticulous evaluation and constructive feedback. We deeply value the insights provided, which have offered us the opportunity to enhance the quality and clarity of our work. In this response, we have systematically addressed each of the concerns raised and have made corresponding revisions to our manuscript where necessary.
>
> We've carefully reviewed each query and concern and have diligently addressed them in our response.
>
> **Reviewer's Concern A :Clarity on data gathering from The Whistle**
>
> We regret that our original manuscript did not clearly explain how the data from "The Whistle" was gathered. We have revised the paper to provide a more detailed explanation.
>
> We didn't simply scrape data from "The Whistle." Instead, we formed a direct collaboration. Our partnership ensured the integrity of data and alignment with scientific research standards. "The Whistle" made necessary adjustments during collection and annotation. As highlighted in the "HeTrue: a New Benchmark Dataset for Hebrew Credibility Assessment" section, their journalists manually curated the HeTrue dataset over years. Each claim was annotated for metadata, semantic features (see full list of features in Appendix), and its full context. Claims were also refined by "The Whistle" for user clarity (“Edited claim”).
>
> For the claim credibility score, we co-designed with "The Whistle" a framework ensuring compatibility with both scientific and journalism standards. The annotation process can be summarized as:
>
> * **Multi-level Review:** As outlined in our paper, we used a rigorous three-stage inter-annotator agreement procedure. Statements were examined by two journalists. Discrepancies prompted a third journalist's review, curbing individual bias
>
> * **Diverse Annotation Team:** Journalists from varied backgrounds ensured factual and unbiased labels
>
> * **IFCN Guidelines:** The participating journalists strictly follow the IFCN guidelines, which uphold transparency, impartiality, and fairness in fact-checking.
>
> * **Nuanced Labeling:** We introduced a detailed 5-point truthfulness scale, complemented by 'Unverifiable' and 'Other' categories. This granularity captures the complexities of real-world statements, avoiding potential bias and oversimplification.
>
> * **Comparison with Established Methods:** Our labeling process aligns with methods employed in recent fact-checking studies, highlighting its consistency with prevailing community standards.
>
> * **Transparent Scoring Rationale:** To further reduce bias and ensure transparency, every claim's rationale for scoring is public and documented.
>
> Moreover, the dataset is released in conjunction with "The Whistle," ensuring that users and researchers have access to both the original source and our refined data for comprehensive insights.
>
> Though our rigorous process yielded a smaller dataset, it ensures the highest quality and reliability.
>
> Lastly, all text-based features were double-checked for accurate transcription. The semantic features benefited from a two-tier verification process involving both a "The Whistle" journalist and an editor. We trust this elucidation sufficiently addresses your concerns.
>
>
>
> **Reviewer's Concern B:** The reviewer emphasized the need for a more focused related work section, highlighting similar datasets for non-English or underrepresented languages. Additionally, they indicated that context and evidence-based methods were not detailed adequately. A recommendation was also made to remove generic parts concerning network analysis for fact-checking, as it didn't align closely with the current research.
>
> **Our Response**
> We truly appreciate the reviewer's suggestions regarding the related work section. We have undertaken the following improvements:
>
> * **Datasets for Underrepresented Languages:** While English remains the predominant language for fake news detection datasets, other languages are often underrepresented. As observed, most efforts to date, such as those by Vlachos and Riedel (2014) and Wang (2017), have extracted real-world claims from dedicated English-based websites like Politifact.
>
> However, Gupta and Srikumar (2021) are notable exceptions, having curated claims from 25 languages. Their multilingual baselines, including models like Claim Only, Attention-based Evidence Aggregator, and Augmenting Metadata Model, have established a promising foundation for multilingual fact-checking. Another commendable initiative is by Baly in “Integrating Stance Detection and Fact Checking in a Unified Corpus” (2018), who compiled a dataset of 219 Arabic statements. Both these studies employed Google's evidence retrieval to bolster claim veracity modeling. Yet, their scope appears limited, especially in terms of integrating context and a rich set of metadata/semantic features.
>
> No Hebrew-specific dataset for fact verification or fake news detection existed until our contribution focusing on real-life statements.
>
>
> * **Context-based Methods:** Context plays a pivotal role in understanding and verifying claims. Few datasets like those by Mitra and Gilbert (2015) and Ma et al. (2016) included claim context, often sourcing from related threads. Since an individual post may contain limited context, these works represent each claim by a set of relevant posts, for example, the thread they originate from. Another unique approach is from Ghanem in "FakeFlow: Fake News Detection by Modeling the Flow of Affective Information," which considers an entire article's segments for information flow. Building on the hypothesis that text interplay is vital, our work is pioneering in presenting sentence-level claims made by public figures, complemented by context.
>
> While our contribution heralds a significant leap in Hebrew-centric datasets for fact verification, it's not confined to that. As our wide-range of experiments and the introduction of empirical evaluation of novel models tailored explicitly for this task: Context-Based Model (CBM), Attention-based Evidence Aggregator (Attn-AE) and Context and Evidence Model (CCEM).
>
> A comprehensive comparison of our dataset against previous works can be found in Table 4 in the appendix.
>
> We hope that these revisions address the reviewer's concerns, and we genuinely thank them for their insightful feedback, which has been instrumental in refining our work.
>
>
>
> **Reviewer's Concern C:**
>  I would have liked to see some simpler baselines for using context and evidence, e.g. just concatenating the two with a <SEP> token or another special token. Have you tried any of these simpler experiments? This would justify that the proposed Attn-EA, CBM, CCEM models add value compared to a simpler method.
>
> **Our Response**
> Thank you for highlighting the importance of foundational methodologies. Due to space constraints and our emphasis on advanced models like Attn-EA, CBM, and CCEM, we limited our discussion on simpler methods. However, post-review, we recognize the value of elaborating on these methods, and we will include them in the final version:
>
> Beyond the Attn-EA, CBM, and CCEM models, we have ventured into more simpler architectures for our context-based models. Specifically, we've also explored:
> * **“Linguistic + Metadata Model”:** This model factors in linguistic intricacies and metadata related to context and evidence. The linguistic features were carefully designed based on past works.
> * **“RNN + FastText:** This approach encapsulated the context, utilizing the power RNN with FastText embeddings.
> The results of which are presented in Table 2.
>
> Regarding simpler methods for amalgamating both evidence and context, our experiments encompassed:
>
> **TF-IDF Vectorization**: Applied TF-IDF to represent the significance of terms in context and evidence. Performance metric: F1 score of 26.2 ± 1.6
> **Direct Concatenation:** Merged context and evidence without the introduction of special tokens. Performance metric: F1 score of 34.3 ± 1.4
> **Sentence Embeddings with a <SEP> Token:** Implemented fixed-size embeddings for the combined representation. Performance metric: F1 score of 44.3 ± 2.4
>
> We trust that these additional insights and experiments, clarify the nuances and value of our proposed models in contrast to the foundational methods. We're genuinely grateful to the reviewer for pointing out this aspect, as it has enabled us to present a more rounded perspective on our research.
>
>
>
> **Reviewer's Concern D:**
>  Can you provide an example of statement, evidence and context in the Appendix?
>
> **Our Response**
> In response to the reviewer request for a specific example of a statement, evidence, and context from the dataset, we provide the following example translated from Hebrew to English:
>
> Original Claim: "You should know that in recent years the parole boards have given fewer and fewer people a third, only about 25% of the people received a third, and as part of our solution, the High Court of Justice said that the living conditions of the prisoners should be increased, and therefore the number of prisoners should also, among other things, be reduced."
>
> Edited Claim: "In recent years, the parole boards have given fewer and fewer people a third, only about 25% of the people received a third."
>
> Evidence Snippet: "The review of the characteristics of the cases and the decisions that were made opens for the first time a window into the work of the parole boards in Israel, which allows for an in-depth study of the results of the proceedings and the reasons that guide the..."
>
> Claim Full Context: "I told both the Ministry of Justice and the media that when the parole board gives a third, the criteria should be good and dangerous behavior, and there is no need to invent all kinds of other criteria that are not in the law [...] You should know that in recent years the parole boards have given fewer and fewer people a third, only about 25% of the people received a third, and as part of our solution, the High Court of Justice said that the living conditions of the prisoners should be increased, and therefore the number of prisoners should also, among other things, be reduced."
>
>
> We hope this example provides clarity on the structure and depth of the dataset entries.
>
> The example in the paper currently excludes this evidence snippet, and we acknowledge the oversight. We will ensure that the "Evidence Snippet" section is incorporated in the revised version of the paper for better clarity and comprehensiveness.
>
> Thank you for your valuable feedback, which will aid in enhancing the quality of the paper.
>
>
> **Reviewer's Concern E:**
> How do you know that the context and evidence do not contain any information about the actual credibility label for the statement? Are these manually inspected?
>
>
> **Our Response**
>
> In response to your query regarding the potential inclusion of credibility labels within the context and evidence:
>
> We have rigorously manually inspected the evidence to ensure the absence of any explicit credibility labels. Through this meticulous process, we identified and addressed instances with problematic evidence. To systematize this and ensure reproducibility without intensive manual oversight, we adopted a two-step procedure(also mentioned in the paper)
>
> 1.Excluded Websites List: As articulated in the manuscript, to mitigate possible biases, we established an 'Excluded Websites' list. This list filters out sources which might inadvertently reveal the claim's veracity, including the original claim source and websites like "The Whistle".
>
> 2.Content Overlap Restriction: For added assurance, any snippets with content overlap exceeding 85% of the original claim are systematically discarded.
>
> It's crucial to emphasize that explicit references to the credibility level are largely confined to "The Whistle" or citations thereof. This makes the exclusion process straightforward. Moreover, other sources that implicitly discuss the claim's credibility — identified by their content overlap with the statement — are also eliminated.
>
> In light of your feedback, we will be sure to emphasize these points more prominently in the final version of our manuscript. We hope this provides clarity on the steps we've taken to maintain the purity and impartiality of our dataset, ensuring its value and reliability for research.
>
>
>
> **Reviewer's Concern F:**
> How were the statements in the HeTrue dataset selected?
>
> **Our Response**
>
> Thank you for your inquiry regarding the HeTrue dataset's statement selection process. The statements in the HeTrue dataset were principally curated by professional journalists from "The Whistle." Their selection was based on a myriad of criteria, encompassing the statement's potential societal impact, its alignment with current events, and the prominence of the statement's author. To ensure impartiality and objectivity in our dataset, we, in close collaboration with “The Whistle”, diligently worked to guarantee that the dataset was free from biases (explained in detail above). We emphasized a balanced representation across factors such as political efficacy, gender, party allegiance, among others, to uphold the dataset's integrity.
>
>
>
> **Reproducibility**
>
> We sincerely appreciate your feedback regarding reproducibility concerns. The integrity and transparency of our research is of utmost importance, and we understand the value of ensuring our results can be easily replicated by the community. To address the concern you raised:
>
> It is important to emphasise that both the dataset and the codebase will be made publicly available upon acceptance of the paper.
>
>
> For all reported experimental results:
>
> We have evaluated all our models across four dataset setups, detailed in Section 3. The experimental setup, encompassing both evaluation and training, is elaborated upon in Section 4.
>
> **Strong Linguistic Baseline:** Various models were examined, with Xgboost being selected based on hyperparameter optimization, as delineated in Appendix A.5. The feature extraction process involves text pre-processing steps, exhaustively described in Appendix A.1 and A.2. We extracted features from both the statement and its encompassing context.
>
> **Metadata-based Models:** Several models based on metadata were utilized. All feature extraction processes, inclusive of pre-processing, are meticulously detailed in Appendix A.4.
>
> **Deep Learning Models:** We have approached experiments in two distinct setups: Statement-Focus and Context-Aware. For each:
>
> - **Statement-Focus Models:** Three models were employed. This included the RNN with Static Word Embeddings utilizing fastText embeddings, selected for its exemplary performance post hyperparameter optimization (detailed in the appendix). Next is the AlephBERT model, a Hebrew BERT variant fortified with a Transformer architecture. It underwent pre-training on 17.6G of Hebrew text and fine-tuning for our specific task. The Attention-based Evidence Aggregator (Attn-EA) model leverages AlephBERT for encoding both claims and evidence, utilizing dot-product attention, as outlined in Section 4.4 (Full formula can be found there too).
>
> - **Context-Aware Models:** Our experiments encompassed the RNN with FastText and Full Context. The Context-Based Model (CBM) and the Combined Context and Evidence Model (CCEM) both employ AlephBERT. The latter integrates statement, context, and external evidence from Google search snippets for assessment. Comprehensive formulas are available in Section 4.4. Complete parameter settings are listed in Appendix A.5.
>
> **Source code** will be made publicly accessible upon acceptance, accompanied by precise specifications and dependencies.
>
> **Evaluation Measure/Statistics:** Metrics, including the macro F1 score, are elucidated in tables 1 and 2 of Section 4. We employ the bootstrap technique to ensure statistical robustness. Comprehensive details are presented in Section 4.2.
>
>
>
> **Regarding the dataset:**
>
> - **Dataset Statistics:** HeTrue encompasses 1021 statements with label distribution outlined in Table 3.
> - **Train/Validation/Test Splits:** We leveraged a nested cross-validation strategy with 5-folds in both inner and outer loops. The detailed approach and methodologies are documented in the appendix and Section 4.
> - **Data Exclusions and Pre-processing:** Only the labels "Unverifiable" and "Other" were omitted. All remaining pre-processing phases are exhaustively detailed in Section 3 and the appendix.
> - **Language:** The dataset is entirely in Hebrew.
> - **Dataset Link:** This will be shared upon paper acceptance.
> - **Data Collection Process:** Detailed in Section 3, including our collaboration with "The Whistle" and the three-stage inter-annotator agreement process.
>
> For all models, we used the publicly available base checkpoints on Huggingface, with Pytorch 2.0.
> All hyperparameters details for our models, described thoroughly in the appendix A.5. We have used Optuna library.
>
>
>
> We hope this comprehensive response should adequately address reproducibility concerns.
>
>
> –
>
> *We're grateful for you bringing to our attention the typos and grammatical mistakes; they have been corrected in the final version of the manuscript.
>
> *All references cited in this response can be found in the bibliography section of our paper.
>
> We deeply appreciate the time and dedication the reviewer has committed to evaluating our manuscript. We have endeavored to address each concern with thoroughness and rigor. It is our sincere hope that, upon reviewing our revisions and the effort we have put into addressing the feedback, the reviewer finds our submission to be significantly enhanced and more robust. Thank you once again for your invaluable insights.

---

### Meta-Review · Area_Chair_xj4g · 2023-09-26

**Recommendation:** 3

**Metareview:**

First of all, thanks to the authors for working on "fact-checking" in a low-resourced language Hebrew. The problem itself is super important in the current scenario. The reviewers pointed out some shortcomings of the paper, which the authors also duly acknowledged. These include a detailed discussion on data collection, including recent relevant works on low-resourced languages, exemplifying the problem definition with relevant artifacts from the dataset, including a few simple baselines, etc., which I believe can be done easily. I would also request the authors to polish the paper in terms of language (reduce verbosity) and, if space permits, move critical discussions from the rebuttals/appendix to the main manuscript. I think the paper has most of the desired information for a resource paper, and the fact-checking dataset on Hebrew is a novel resource. I hope the authors would take the suggestions and cues from the reviewers to improve their paper.

---

### Decision · Program_Chairs · 2023-10-07

**Decision:**

Accept-Findings

**Comment:**

First of all, thanks to the authors for working on "fact-checking" in a low-resourced language Hebrew. The problem itself is super important in the current scenario. The reviewers pointed out some shortcomings of the paper, which the authors also duly acknowledged. These include a detailed discussion on data collection, including recent relevant works on low-resourced languages, exemplifying the problem definition with relevant artifacts from the dataset, including a few simple baselines, etc., which I believe can be done easily. I would also request the authors to polish the paper in terms of language (reduce verbosity) and, if space permits, move critical discussions from the rebuttals/appendix to the main manuscript. I think the paper has most of the desired information for a resource paper, and the fact-checking dataset on Hebrew is a novel resource. I hope the authors would take the suggestions and cues from the reviewers to improve their paper.